# Magnetic Stirling Cycle for Qubits with Anisotropy near the Quantum Critical Point

Cristóbal Araya [1], Francisco J. Peña [2,3,*], Ariel Norambuena [4], Bastián Castorene [2] and Patricio Vargas [1,*]

[1] Departamento de Física, CEDENNA, Universidad Técnica Federico Santa María, Casilla 110-V, Valparaíso 2390123, Chile; cristobal.arayal@sansano.usm.cl

[2] Departamento de Física, Universidad Técnica Federico Santa María, Casilla 110-V, Valparaíso 2390123, Chile; bastian.castorene@usm.cl

[3] Millennium Nucleus in NanoBioPhysics (NNBP), Av. España 1680, Valparaíso 11520, Chile

[4] Centro Multidisciplinario de Física, Universidad Mayor, Camino la Pirámide 5750, Huechuraba, Santiago 7500994, Chile; ariel.norambuena@umayor.cl

[*] Correspondence: francisco.penar@usm.cl (F.J.P.); patricio.vargas@usm.cl (P.V.)

**Abstract:** We studied the performance of a quantum magnetic Stirling cycle that uses a working substance composed of two entangled antiferromagnetic qubits ($J$) under the influence of an external magnetic field ($B_z$) and an uniaxial anisotropy field ($K$) along the total spin in the y-direction. The efficiency and work were calculated as a function of $B_z$ and for different values of the anisotropy constant $K$ given hot and cold reservoir temperatures. The anisotropy has been shown to extend the region of the external magnetic field in which the Stirling cycle is more efficient compared to the ideal case.

**Keywords:** entangled qubits, magnetic cycle; quantum thermodynamics





## 1. Introduction

Quantum heat engines (QHEs) are currently the subject of very active investigation, which aims to discover and develop highly efficient nanoscale devices that function with quantum working substances. These devices are also characterized by the thermodynamic cycle of operation and the dynamics governing the cycle [1–23].

When applied to quantum thermodynamics with an external magnetic field as the drive parameter, the Stirling cycle comprises two isothermal strokes and two isomagnetic strokes. In the isothermal strokes, the working substance interacts with a thermal reservoir at varying temperatures, whereas in the isomagnetic strokes, the magnetic field affecting the system remains constant. The efficiency of this cycle is notably influenced by the intensity of the external magnetic field, as it directly impacts the energy spectrum and states of the working substance. Consequently, the Stirling cycle in quantum thermodynamics, employing an external magnetic field as the drive parameter, exhibits promising potential for applications in magnetic refrigeration and related fields [24–26].

Recently, a quantum Stirling cycle based on two coupled spins near a quantum critical point (QCP) was examined, the XX isotropic Heisenberg model, with a magnetic field along the z-direction [24]. The researchers investigated the system's quantum phase transition, entanglement, and correlation, showing that by choosing specific cycle parameters, the system can operate as a heat engine or refrigerator over a wide range of magnetic fields and low temperatures. Whereas the Stirling heat engine can reach Carnot efficiency near the critical point for high magnetic fields, the refrigerator cycle can approach the Carnot limit for low magnetic fields. However, at higher temperatures, the system's performance deviates significantly from the Carnot limits, and maximum work output does not necessarily correspond to maximum efficiency [24].

Exploring a quantum heat engine using the Heisenberg XX system with three qubits was studied [27], seeking for efficiency optimization. However, it does not equal Carnot's efficiency when the external field is small enough.

A recent study, Kuznetsova et al. [28], examines a two-qubit Heisenberg XYZ model with DM (Dzyaloshinskii–Moriya) and KSEA (Kaplan–Shekhtman–Entin-Wohlman–Aharony) interactions under a nonuniform external magnetic field as a working substance for a quantum Otto thermal machine. The analysis shows that the role of DM and KSEA interactions changes as the longitudinal exchange constant alters the system's behavior from antiferromagnetic to ferromagnetic. Different operating modes of the thermal machine are identified, including heat engine, refrigerator, heater or dissipator, and thermal accelerator or cold-bath heater.

Another publication [29] explores the influence of anisotropy in the exchange constants and magnetic field on entanglement in a two-qubit XY model. The study reveals that the combined effect of the anisotropy parameter and the magnetic field can generate entanglement in regions of the parameter space where it was absent in the isotropic case. This suggests the possibility of controlling and producing entanglement in two-spin systems even at finite temperatures.

In this study, based on the effect of anisotropy in extending the range of entanglement, we present an investigation of a Stirling cycle utilizing a coupled system of two qubits subjected to an external magnetic field along the z-direction and influenced by uniaxial anisotropy energy along the total spin in the y-direction. Our focus lies in analyzing the efficiency and work of the magnetic Stirling cycle. Specifically, we examine the impact of anisotropy on the temperature range where the efficiency achieves its peak value, the extent of the magnetic field interval where high efficiency is observed, and how it is affected by anisotropy.

## 2. Model

The working medium consists of two spin-1/2 particles described by the Heisenberg XX model with an uniaxial anisotropy along the $y$ direction. The Hamiltonian of the system is given by:

$$H = \frac{J}{2}(\sigma_1^x \sigma_2^x + \sigma_1^y \sigma_2^y) + \frac{B}{2}(\sigma_1^z + \sigma_2^z) + \frac{K}{2}(\sigma_1^y + \sigma_2^y)^2,$$ (1)

where $\sigma_i^\alpha (i = 1, 2; \alpha = x, y, z)$ are the Pauli matrices, $J$ is the exchange coupling constant, $B$ is the external field along the $z$ axis, and $K$ is the anisotropy term. This Hamiltonian can be written in a matrix form as:

$$H = \begin{pmatrix} B+K & 0 & 0 & -K \\ 0 & K & J+K & 0 \\ 0 & J+K & K & 0 \\ -K & 0 & 0 & K-B \end{pmatrix}.$$ (2)

The eigenvalues and the corresponding eigenvectors in the two-qubit computational basis are defined as:

$$E_1 = \bar{E} - J_{\text{eff}}; \qquad |\psi_1\rangle = \frac{1}{\sqrt{2}}(|01\rangle - |10\rangle),$$ (3)

$$E_2 = \bar{E} + J_{\text{eff}}; \qquad |\psi_2\rangle = \frac{1}{\sqrt{2}}(|01\rangle + |10\rangle),$$ (4)

$$E_3 = \bar{E} - B_{\text{eff}}; \qquad |\psi_3\rangle = \frac{1}{\sqrt{N_-^2 + 1}}[|00\rangle - N_- |11\rangle],$$ (5)

$$E_4 = \bar{E} + B_{\text{eff}}; \qquad |\psi_4\rangle = \frac{1}{\sqrt{N_+^2 + 1}}[|00\rangle - N_+ |11\rangle],$$ (6)

where $\bar{E} = \sum_{i=1}^{4} E_i/4 = K$ is the average energy, $J_{\mathrm{eff}} = J + K$ is the effective exchange coupling constant, $B_{\mathrm{eff}} = \sqrt{B^2 + K^2}$ is the effective magnetic field, and $N_{\pm} = (B \pm B_{\mathrm{eff}})/K$. First, we note that eigenstates $|\psi_1\rangle$ and $|\psi_2\rangle$ are two Bell states that are robust against magnetic field fluctuations. Therefore, only states $|\psi_3\rangle$ and $|\psi_4\rangle$ are affected by the magnetic field, leading to a crossing point ($E_1 = E_3$ and $E_2 = E_4$) at the critical magnetic field $B_{\mathrm{crit}} = \sqrt{J_{\mathrm{eff}}^2 - K^2} = \sqrt{J^2 + 2JK}$. This critical magnetic field only exists if $K \geq -J/2$. Otherwise, there is no crossing point between energy levels. The existence of this critical magnetic field defines the QCP, where the ground state is changed from $|\psi_1\rangle$ to $|\psi_3\rangle$ when the magnetic field changes from $B < B_{\mathrm{crit}}$ to $B > B_{\mathrm{crit}}$. In addition, for $K = -J$ ($J_{\mathrm{eff}} = 0$), the states $|\psi_1\rangle$ and $|\psi_2\rangle$ are degenerated.

To have more intuition about our working medium, we plot the energy levels in Figure 1. Different anisotropy values given by $K = 0.5$ (Figure 1a) and $K = -0.4$ (Figure 1b) are chosen to illustrate the role of the anisotropy. Briefly, the anisotropy introduces two effects, namely, the change in the average energy and the location of the QCP. From the expression $B_{\mathrm{crit}} = \sqrt{J^2 + 2JK}$, we note that the QCP is displaced to the right (left) for positive (negative) values of $K$, as shown in Figure 1a,b. Positive values of $K$ widen the gap between the energy levels, whereas negative values cause them to get closer. This will affect the population of the energy levels when the Stirling cycle is performed, as will be discussed later.

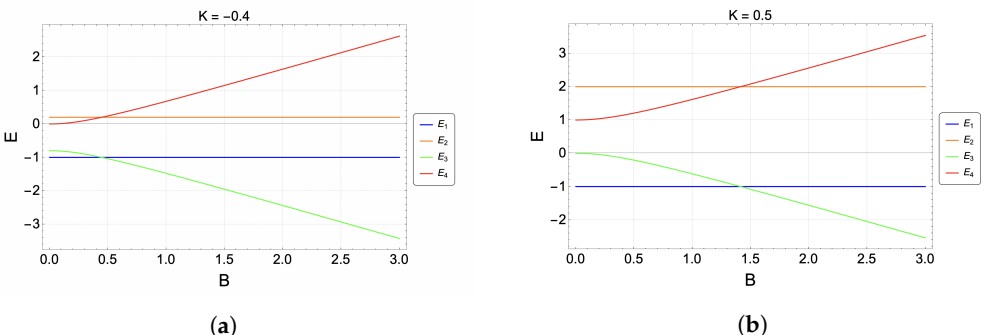

(a)  (b)

**Figure 1.** Variation of the energy levels for (**a**) $K = -0.4$, (**b**) $K = 0.5$; $J = 1$ for both figures.

The calculations to be presented for the Stirling cycle consider that, at any instant, the thermodynamic quantities are fully defined from the canonical partition function of the system ($k_B = 1$):

$$Z = \sum_{i=1}^{4} \exp\left(-E_i/T\right) = 2e^{-\frac{\bar{E}}{T}}\left(\cosh\left(\frac{B_{\mathrm{eff}}}{T}\right) + \cosh\left(\frac{J_{\mathrm{eff}}}{T}\right)\right). \quad (7)$$

Consequently, the free energy $F$, the internal energy $U$, entropy $S$, magnetization $M$, and heat capacity $C$ of the system can be determined through the formulas (using $k_B = 1$):

$$F = -T \ln Z, \qquad U = T^2 \frac{\partial \ln Z}{\partial T}, \qquad S = -\frac{\partial F}{\partial T}, \qquad M = -\frac{\partial F}{\partial B}, \qquad C = \frac{\partial U}{\partial T}. \quad (8)$$

The populations at thermal equilibrium are defined as $P_i = \exp\left(-E_i/T\right)/Z$ (with $i = 1, 2, 3, 4$), which allow us to introduce the density matrix

$$\rho = \sum_{i=1}^{4} P_i |\psi_i\rangle \langle\psi_i|. \quad (9)$$

Note that the density matrix of the system describes a four-level system (two interacting qubits) such that $\mathrm{Tr}(\rho) = 1$ and $\rho = \rho^{\dagger}$. In the presence of exchange ($J$) and anisotropy ($K$), both qubits experience an interaction that is the entanglement source. Thus, in the next section, we will analyze the role of correlation in the thermodynamical context.

## 3. Spin Correlations and Quantum Phase Transition

To explain the role of correlations, we shall discuss how spin correlations and quantum correlations are connected to the quantum phase transition (QPT). In particular, the main idea is to connect correlations with QCP and their relevance to the quantum Stirling cycle. The classification and detail of a quantum phase transition have been extensively studied with different approaches [30,31]. Here, we analyze two relevant quantities: two-point spin correlations and concurrence close to the QCP.

### 3.1. Spin Correlations

Spin correlations are relevant when many-body systems exhibit short- and long-range interactions [32]. When $K = 0$ (zero anisotropy) [24], the QPT is expected to be observed in both spin correlations and concurrence as the ground state changes from a partially entangled to a maximally entangled state (Bell state) at low temperatures. To observe this effect, we introduce the two-point correlation function:

$$C_{ij}^{\alpha\beta} = \langle \sigma_i^\alpha \sigma_j^\beta \rangle - \langle \sigma_i^\alpha \rangle \langle \sigma_j^\beta \rangle. \tag{10}$$

The above spin correlation function is calculated using the density matrix at thermal equilibrium, and then $\langle \sigma_i^\alpha \rangle = \mathrm{Tr}(\sigma_i^\alpha \rho)$, where $\rho$ is defined in Equation (9). Here, $\{i, j\} \in \{1, 2\}$ are the site labels and $\{\alpha, \beta\} \in \{x, y, z\}$ are the spin components. Let us suppose that the two-qubit system is described by the product (uncorrelated) state $\rho = \rho_1 \otimes \rho_2$. In this case, a simple calculation tells us that $C_{12}^{\alpha\beta} = 0 \; \forall \; \alpha, \beta$, and, then, correlations (along any axis) are not present. Thus, any value $C_{12}^{\alpha\beta} \neq 0$ tells us that the system has spin correlations.

For the particular case of two qubits, we have indistinguishable particles, and, then, $C_{12}^{\alpha\beta} = C_{21}^{\alpha\beta}$ and $C_{12}^{\alpha\beta} = C_{12}^{\beta\alpha}$. This reduces the number of possible calculations of the symmetric tensor $C_{12}^{\alpha\beta}$ to the upper diagonal terms (six elements). Physically, because we have exchange coupling ($J$) and anisotropy ($K$), this naturally leads to a certain degree of spin correlation between the qubits. For the state in thermal equilibrium, we found:

$$C_{12}^{xx} = -\frac{\sinh(\frac{J_{\mathrm{eff}}}{T}) - \frac{K \sinh(\frac{B_{\mathrm{eff}}}{T})}{B_{\mathrm{eff}}}}{\cosh(\frac{J_{\mathrm{eff}}}{T}) + \cosh(\frac{B_{\mathrm{eff}}}{T})}, \tag{11}$$

$$C_{12}^{yy} = -\frac{\sinh(\frac{J_{\mathrm{eff}}}{T}) + \frac{K \sinh(\frac{B_{\mathrm{eff}}}{T})}{B_{\mathrm{eff}}}}{\cosh(\frac{J_{\mathrm{eff}}}{T}) + \cosh(\frac{B_{\mathrm{eff}}}{T})}, \tag{12}$$

$$C_{12}^{zz} = \frac{B^2 - B_{\mathrm{eff}}^2 \cosh(\frac{2J_{\mathrm{eff}}}{T}) + K^2 \cosh(\frac{2B_{\mathrm{eff}}}{T})}{2B_{\mathrm{eff}}^2 \left( \cosh(\frac{J_{\mathrm{eff}}}{T}) + \cosh(\frac{B_{\mathrm{eff}}}{T}) \right)^2}. \tag{13}$$

Due to symmetry reasons, the other three elements, $C_{12}^{xy}$, $C_{12}^{xz}$ and $C_{12}^{yz}$, of the correlation tensor vanish [30,33,34]. Figure 2 shows the xx-correlation as a function of $B$ and $T$. As in Ref. [24], the correlation abruptly changes around the critical point when $T \to 0$. For example, in Figure 2b, with $J = 1$ and $K = 0.5$, the location of the non-analytical behavior of the spin correlation $C_{12}^{xx}$ at $T = 0$ is exactly the critical magnetic field $B_{\mathrm{crit}} = \sqrt{2} \sim 1.4$. When the anisotropy is negative $K = -0.4$ (with $J = 1$), we note in Figure 2a that the critical point is shifted to lower field values. In this particular case, we have a smaller critical field $B_{\mathrm{crit}} = \sqrt{\frac{1}{5}} \sim 0.45$. These results indicate the existence of a first-order QPT [35,36]. We note that the same behavior observed for $C_{12}^{xx}$ around the QCP is also reflected in other spin correlation terms $C_{12}^{\alpha\beta}$.

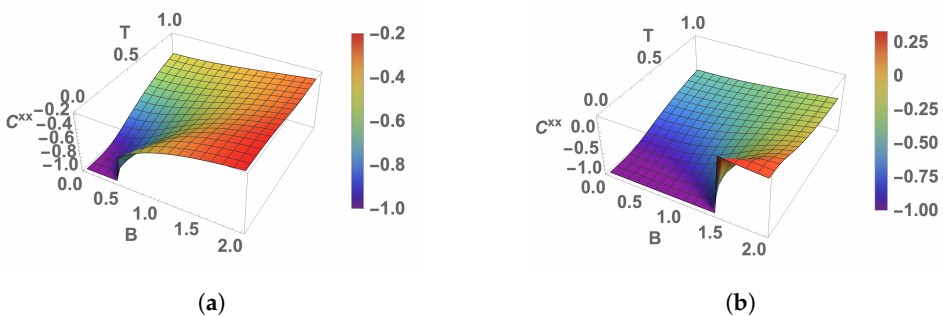

(**a**)                                                                  (**b**)

**Figure 2.** Variation of the xx-correlation function for (**a**) $K = -0.4$ (**b**) $K = 0.5$. We have selected $J = 1$ for these cases.

### 3.2. Quantum Entanglement

It is well-known from information theory that different entanglement measures can be used to identify QPT. In particular, for the concurrence defined by Wootters [37] in the context of a two-qubit system, it is a useful QC measure based on the separability of the system. The concurrence is defined as follows:

$$c = \max\{0, \sqrt{\lambda_1} - \sqrt{\lambda_2} - \sqrt{\lambda_3} - \sqrt{\lambda_4}\}, \tag{14}$$

where $\lambda_j (j = 1, 2, 3, 4)$ are the eigenvalues of the matrix $\rho(\sigma_1^y \otimes \sigma_2^y)\rho^\dagger(\sigma_1^y \otimes \sigma_2^y)$ ordered from the largest to the smallest value. For a two-spin system described by uncorrelated state $\rho = \rho_1 \otimes \rho_2$, we have the theoretical result $c = 0$, as explained in Ref. [38].

In Figure 3, it can be seen that, at low temperatures, the concurrence abruptly changes from a low value to a maximum of one as the ground states transition from the state $|\psi_3\rangle$ to $|\psi_1\rangle$ when the magnetic field decreases past the QCP. Also, with the introduction of anisotropy, a region with almost no entanglement appears. Contrary to the case with $K = 0$, the concurrence is greater than zero for fields higher than the QCP at low temperatures. This happens because the ground state is now a mix of $|11\rangle$ and $|00\rangle$ states, although this behavior disappears for a higher magnetic field due to the ground state beginning to approximate the state $|00\rangle$. The correlation of the spin in all three axes presents a similar behavior around the QCP, where the value changes abruptly at low temperatures.

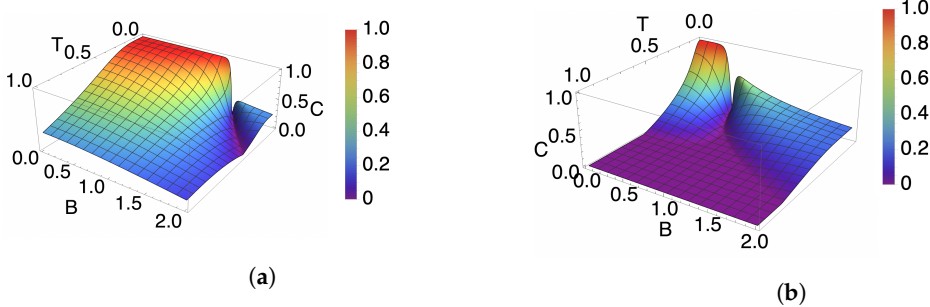

(**a**)                                                                  (**b**)

**Figure 3.** Concurrence as function of temperature and magnetic field with $J = 1$ in both graphics and $K = 0.5$ and $K = -0.4$ for (**a**) and (**b**), respectively.

### 3.3. Linking Correlation with Thermodynamics

Connecting thermodynamic quantities such as entropy and free energy with correlations derived from statistical information theory is intriguing. In this direction, for instance, in Ref. [39], the authors explicitly connect concurrence and thermodynamical quantities such as energy and work at zero temperature. More formally, some authors use relative entropy to build a bridge between information theory and thermodynamics [40–42]. Here, we are interested in giving a more intuitive relation that exhibits the role of anisotropy in the spin correlations.

To better understand the origin of spin correlations and their connection with thermodynamics, let us consider the system Hamiltonian given in Equation (1). An alternative (but equivalent) expression for the system Hamiltonian is:

$$H = \mathbb{1}K + \frac{(J_{\text{eff}} - K)}{2}\sigma_1^x\sigma_2^x + \frac{(J_{\text{eff}} + K)}{2}\sigma_1^y\sigma_2^y + \frac{B}{2}(\sigma_1^z + \sigma_2^z),  \quad (15)$$

where $\mathbb{1}$ is the identity matrix for the Hilbert space of the qubits. From Equation (15), we observe that the anisotropy plays a role in differentiating correlations around the $x$ and $y$ axes. In fact, for $K = 0$, the anisotropy effect disappears, and one expects to have the same spin correlations for $C_{12}^{xx}$ and $C_{12}^{yy}$. Starting from the free energy function, $F = K - T\ln(2(\cosh(J_{\text{eff}}/T) + \cosh(B_{\text{eff}}/T)))$, we obtain the following analytical expressions:

$$C_{12}^{xx,yy} = \left(\frac{\partial}{\partial J_{\text{eff}}} \mp \frac{K}{B_{\text{eff}}}\frac{\partial}{\partial B_{\text{eff}}}\right)F, \quad C_{12}^{zz} = -\left(\frac{\partial F}{\partial J_{\text{eff}}}\right)^2 - \left(\frac{K}{B_{\text{eff}}}\right)^2\left[2e^{(F-K)/T} + \left(\frac{\partial F}{\partial B_{\text{eff}}}\right)^2\right], \quad (16)$$

where $C_{12}^{xx}$ ($C_{12}^{yy}$) is obtained using the minus (plus) sign of the differential operator $\partial/\partial J_{\text{eff}} \mp (K/B_{\text{eff}})\partial/\partial B_{\text{eff}}$. When the anisotropy is not present ($K = 0$), we have $J_{\text{eff}} = J$, and the spin correlations reduces to $C_{12}^{xx} = C_{12}^{yy} = \partial F/\partial J$ and $C_{12}^{zz} = -(\partial F/\partial J)^2$, in good agreement with Equations (11)–(13). Most importantly, based on Equation (16), we can construct a direct connection between thermodynamics and spin correlations. The inverse relations, namely, the free energy function ($F$), entropy ($S$), energy ($U$), magnetization ($M$), and specific heat ($C$) as a function of spin correlations, are more challenging. However, for the particular system of two interacting qubits, we found

$$F = K - T\ln(G_1), \quad S = \ln(G_1) + \frac{G_2}{T}, \quad U = K + G_2, \quad M = \frac{B}{2K}C_-, \quad C = \frac{J_{\text{eff}}B_{\text{eff}}^2}{2KT^2}C_-C_+. \quad (17)$$

where $C_\pm = C_{12}^{yy} \pm C_{12}^{xx}$. Here, $G_1$ and $G_2$ are functions that depend on the spin correlations and are given by

$$G_1 = \frac{16K^2}{\sqrt{(B_{\text{eff}}C_-)^4 - 2(KB_{\text{eff}}C_-)^2(4 + C_+^2) + K^4(C_+^2 - 4)}}, \quad (18)$$

$$G_2 = \frac{J_{\text{eff}}}{2}C_+ + \frac{B_{\text{eff}}^2}{2K}C_-. \quad (19)$$

We remark that all these expressions only depend on the spin correlation functions $C_{12}^{xx}$ and $C_{12}^{yy}$ leading to a crucial role of the anisotropy. Moreover, our expressions are also helpful at any temperature, extending previous results at zero temperature. We have all the ingredients to explain our proposal for the magnetic Stirling cycle.

## 4. Quantum Stirling Cycle

Analogous to the classical cycle, the quantum Stirling cycle consists of four strokes: two isothermal and two isomagnetic processes (equivalent to the isochoric processes of the standard case). During the isothermal process, the system remains in thermal equilibrium with one of two thermal baths at temperatures $T = T_H$ (high) or $T = T_L$ (low), with $T_H > T_L$. At the same time, the external magnetic field is varied between $B_H$ and $B_L$ during the process, satisfying the condition $B_L < B_H$. Concerning the isomagnetic trajectories, the coupling with the baths is switched, and the systems carry out a process at a constant magnetic field (at $B_L$ or $B_H$). The cycle can be visualized pictorially in Figure 4 where, as we have mentioned, the control parameter $\lambda$ corresponds to the external magnetic field $B$ acting on the system. The analysis by stage of the cycle described in the Figure 4 can be summarized as follows:

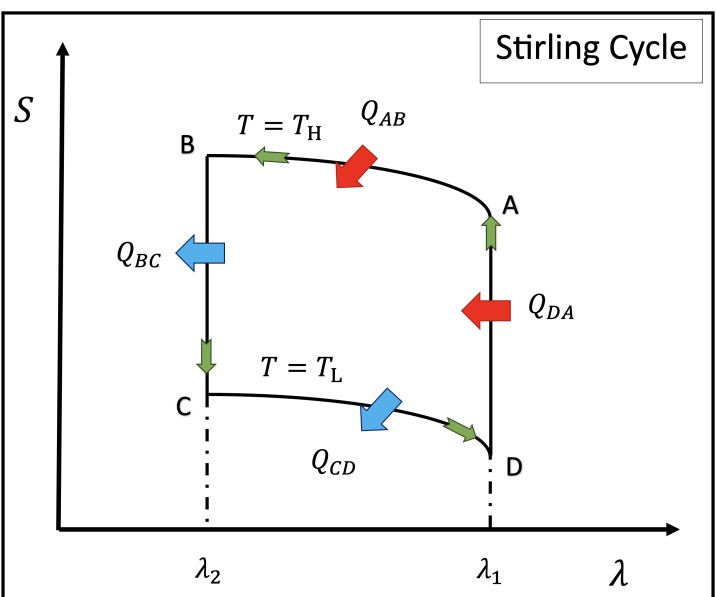

**Figure 4.** Diagram of the Stirling cycle regarding entropy and the control parameter $\lambda$. In our case, $\lambda \to B$ corresponds to the external perpendicular magnetic field value along the $z$-axis.

*Isothermal stages:* These processes correspond to the $A \to B$ and $C \to D$ trajectories of Figure 4. For the first stage $A \to B$ (third stage $C \to D$), the system is held at a fixed temperature $T_H$ ($T_L$), and the entropy changes during $A \to B$ ($C \to D$) due to the variation of the external magnetic field. Consequently, the heat exchange of the system during these strokes can be obtained from the change of entropy along the processes at constant temperature in the form:

$$Q_{AB} = T_H(S_B - S_A), \tag{20}$$
$$Q_{CD} = T_L(S_D - S_C). \tag{21}$$

*Isomagnetic stages:* These processes correspond to the $B \to C$ ($B_L =$ cnt.) and $D \to A$ ($B_H =$ cnt.) trajectories of Figure 4. In these two strokes, there is no work done; therefore, the heat exchange is obtained as the change of internal energy while the system changes from the thermal bath with temperature $T_H$ to the one with temperature $T_L$ during the $B \to C$ stage, and from $T_L$ to $T_H$ during the $D \to A$ stroke. Consequently, the expressions for the heats in these stages are given by:

$$Q_{BC} = U_C - U_B, \tag{22}$$
$$Q_{DA} = U_A - U_D. \tag{23}$$

From the first law of thermodynamics ($dU = \delta Q - \delta W$), and using the fact the variation of $U$ in a closed cycle is zero, the total work $W$ can be obtained as:

$$W = Q_H + Q_L = Q_{AB} + Q_{DA} + Q_{BC} + Q_{CD}, \tag{24}$$

where we define $Q_H = Q_{AB} + Q_{DA}$ and $Q_L = Q_{BC} + Q_{CD}$. The signs of $Q_H$, $Q_L$, and $W$ will define the machine's behavior. Two possible cases stand out: The proposed machine may operate as an engine or refrigerator. The analysis of the signs of heat and total work for each case are summarized in Table 1.

**Table 1.** Heat and total work sign convention for classifying the thermal machine as a heat engine or refrigerator.

| Heat and Work | Engine | Refrigerator |
|:---:|:---:|:---:|
| $Q_H$ | $> 0$ | $< 0$ |
| $Q_L$ | $< 0$ | $> 0$ |
| $W$ | $> 0$ | $> 0$ |
| $\eta$ | $< 1$ | - - - - |
| COP | - - - - | $> 1$ (expected) |

If the thermal machine satisfies the conditions of an engine, we can define the efficiency $\eta$, which corresponds to the ratio between the total work and the heat the system absorbs. Therefore, $\eta$ is given by

$$\eta = \frac{W}{Q_H}. \tag{25}$$

If the machine responds to a refrigerator-type behavior, we can define the coefficient of performance (COP) in an analogy to the efficiency of engines, and it is given by

$$\epsilon = \frac{Q_L}{\mid W \mid}. \tag{26}$$

## 5. Results and Discussion

In this section, we will begin the discussion by focusing on the effects of anisotropy on the proposed cycle's efficiency results. Computational grids in Mathematica software can be found in supplementary materials. Figure 5 shows the efficiency as a function of the parameter $B_L$ for different values of anisotropy: $K = -0.4$ (blue solid line), $K = 0$ (gray dotted line), and $K = 0.5$ (red solid line) with fixed values of $T_H = 0.2$, $T_L = 0.12$, $B_H = 3$, and $J = 1$. We first observe that the maximum efficiency for each case of $K$ is located exactly at the quantum critical point ($B_{\text{crit}} = \sqrt{J^2 + 2JK}$). Consequently, the maximum efficiency moves to the left (right) when $K$ decreases (increases).

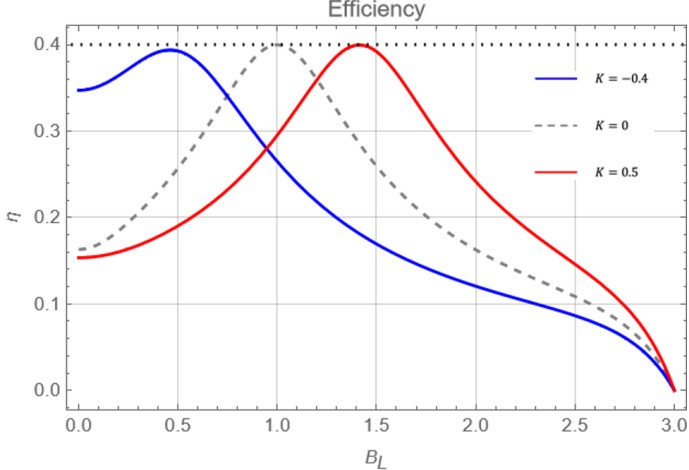

**Figure 5.** Efficiency as a function of $B_L$ for $T_H = 0.2$, $T_L = 0.12$, $B_H = 3$, $J = 1$, and $K = -0.4$, $K = 0$, and $K = 0.5$ for the blue, dashed gray, and red lines, respectively. The black dotted line is the Carnot efficiency for the temperature already given.

An important point to note is that the model recovers the efficiency with $K = 0$ reported in Ref. [24], where, at point $B = J = 1$, it is found to be equal to Carnot. If we observe the red curve in Figure 5, we notice that at the critical point with positive $K$, the maximum efficiency also corresponds to Carnot. However, for negative $K$, this value is not reached and is slightly below. This can be explained by what is mentioned in Ref. [24],

where it is said that the value of the Carnot efficiency will be obtained just at the critical point for very low temperatures of the thermal reservoirs where the fundamental state of energy is mostly populated. As in the case with negative anisotropy that makes the levels come together even more, for the same temperatures presented in Figure 5, the participation of the other levels of energy becomes non-negligible, causing, in consequence, a slight decrease of the efficiency around the critical point evidenced in the blue curve of Figure 5. However, if lower temperature reservoirs were selected, Carnot would again be achieved for negative anisotropies. As in situations without uniaxial anisotropy, the efficiency decreases as $B_L$ moves further away from the critical value. Nonetheless, this decline is not as rapid when $K$ is not equal to zero. Therefore, when the anisotropy is present, the efficiency remains optimal in a broader range for $B_L$.

For $K = 0.5$, the efficiency is above half of Carnot's in a range 9.2% wider than for $K = 0$. It can be noted, also, from Figure 6, that work (green curve for Figure 6a,b) is maximized at the same point of $B_l$ as efficiency. That is, for the case of $K = 0.5$ (Figure 6a), work is maximized at the point $B_l \sim 1.4$, whereas for $K = -0.4$ (Figure 6b), the maximum of work is displayed at $B_l \sim 0.45$.

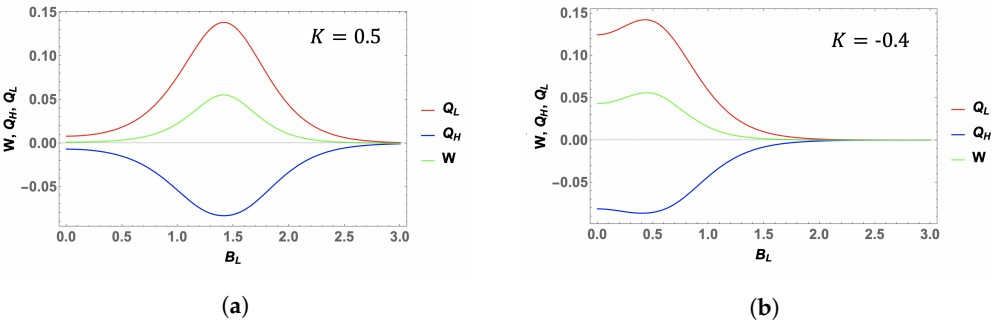

(**a**)                                                                                                                    (**b**)

**Figure 6.** Heat input (red curve), heat output (blue curve), and total work (green curve) for the case of (**a**) $K = 0.5$ and (**b**) $K = -0.4$. The parameters for this plot are : $T_H = 0.2$, $T_L = 0.12$, $B_H = 3$, and $J = 1$.

An interesting result can be seen in Figure 7a, where the efficiency is shown as a function of $T_H$ for $T_L = 0.6T_H$, $B_L = 1$, $B_H = 5$, and $J = 1$ for $K = 0.5$ (red curve), $K = 0$ (gray curve), and $K = -0.4$ (blue curve). We notice that, at the beginning of the graph, the efficiencies are all equal for very small values of $T_H$. Then, they are ordered so that the efficiency for positive $K$ is the highest and negative $K$ the lowest. However, as $T_H$ increases, this behavior is completely reversed, and even in the case of negative $K$, a peak in efficiency is observed again. This does not occur (for the selected parameters) for the case of $K = 0$ (same as Ref. [24]) and for $K > 0$, which have a monotonic decreasing behavior with increasing $T_H$. The reason behind this maximum in the blue curve of Figure 7a seems to be the thermal population of the level $E_1$ and $E_2$ near the temperature of the peak while having a value of $B_H$ high enough to keep the $E_4$ level mostly unoccupied, effectively working similarly to a three levels system. This peak would also appear in the case of positive $K$ but would require higher fields ($B_H > 5$ for this example) and higher temperatures $T_H$ to have the same effect as for $K < 0$. Finally, in Figure 7b, we show the work as a function of $T_H$ for the same fixed parameters mentioned for panel (a). We note that the work is much higher for negative anisotropy values over the range of $T_H$ shown, whereas the lowest useful work is in the positive K case. From here, it can be concluded that the inclusion of anisotropy for values of $T_H > 0.8$ presents the best performance in efficiency and work of the proposed cycle.

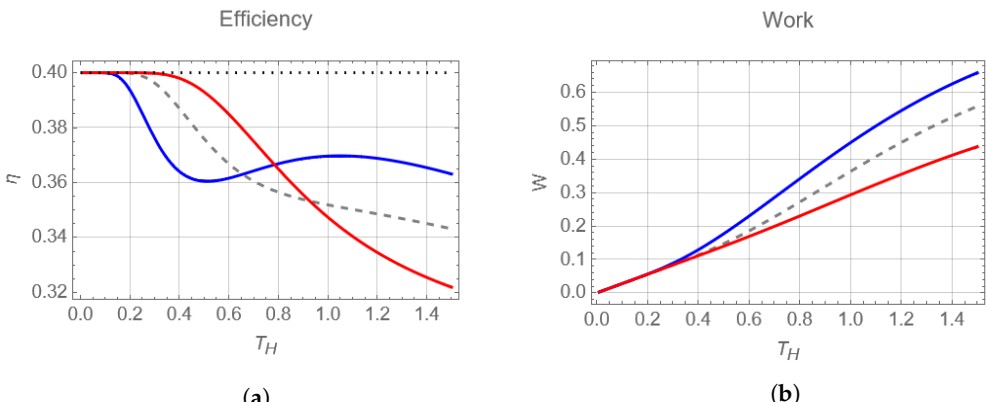

**Figure 7.** Efficiency (**a**) and work (**b**) as function of $T_H$ (maintaining $T_L = 0.6T_H$) at QCP for $B_L = 1$, with $K = -0.4$, $K = 0$, and $K = 0.5$ for the blue, gray dashed, and red lines, respectively. The other parameters are $J = 1$ and $B_H = 5$. The black dotted line is the Carnot efficiency when $T_L = 0.6T_H$.

## 6. Conclusions

In this work, we have analyzed the performance of a Stirling engine whose working substance corresponds to an interacting system of two ½ spin particles described through a Heisenberg XX-type model with a uniaxial anisotropy with the total spin along the y-axis direction. The control parameter of the proposed cycle corresponds to an external field perpendicular to the system. Correlation and concurrence have analyzed the model's quantum critical points (QCP). It has been shown that these QCPs play a fundamental role in the work and efficiency of the proposed cycle [24,27–29]. Our results indicate that incorporating anisotropy (which appears explicitly in the QCPs) runs the maximum efficiency (Carnot) points in the cycle control parameter space either to the left or the right of the previously reported case for $K = 0$ (without anisotropy), showing that for $K < 0$ the maximum possible efficiency is obtained for much lower fields than with $K = 0$. In addition, due to the structure of the energy levels, it is possible to find areas where the incorporation of anisotropy shows better thermal performance than in its absence, as well as better overall work. Therefore, incorporating the anisotropic term improves the machine's performance from any thermodynamic point of view. Furthermore, we successfully demonstrated the close interrelationship between thermodynamics and information theory in this bipartite quantum system. This was achieved by explicitly deriving all thermodynamic quantities for the system through their expression in terms of spin correlation functions.

**Supplementary Materials:** The following supporting information can be downloaded at: https://www.mdpi.com/article/10.3390/technologies11060169/s1.

**Author Contributions:** C.A. and P.V. conceived the idea and formulated the theory. C.A., P.V. and F.J.P. contributed to discussions during the entire work and editing of the manuscript. A.N. and B.C. contributed to connecting correlation with thermodynamics. All authors have read and approved the final manuscript.

**Funding:** This research was funding by Financiamiento Basal para Centros Científicos y Tecnológicos de Excelencia, under Project AFB 220001 (Chile), FONDECYT grant 1210312, ANID Fondecyt Iniciación en Investigación 2020 grant no. 11200032 and "Millennium Nucleus in NanoBioPhysics" project NNBP NCN2021_021.

**Data Availability Statement:** The data presented in this study are available on request from the corresponding author.

**Acknowledgments:** F.J.P. acknowledges DGIIE from Santa Maria University. A.N. acknowledges financial support from Fondecyt Iniciación No. 11220266. The authors acknowledge DTI-USM for using "Mathematica Online Unlimited Site" at the Universidad Técnica Federico Santa María.

**Conflicts of Interest:** The authors declare no conflict of interest.

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
