# Peer review of "Magnetic Stirling Cycle for Qubits with Anisotropy near the Quantum Critical Point"

_technologies, doi:10.3390/technologies11060169_

Round 1

Reviewer 1 Report

Comments and Suggestions for Authors

In this paper, the authors proposed to a scheme to the investigation of a Stirling cycle utilizing a coupled system of two qubits subjected to an external magnetic field along the z-direction and influenced by uniaxial anisotropy energy along the total spin in the y-direction. Meanwhile, they examined the impact of anisotropy on the temperature range where the efficiency achieves its peak value. The paper is interesting and reasonable. I suggest that this paper be published on Technologies, but the following issues need to be dealt with before that:

1. The Section 3, “Correlations and quantum phase transition” should be changed into “Spin correlations and quantum phase transition”.

2. In line 116, these results indicate the existence of a first-order QPT. How to define first-order QPT, second-order, and others?

3. In Section 3.2, the authors studied the relation between entanglement measure and QPT. Maybe it is more proper to change the title “Quantum correlations” into “Quantum entanglement”

4. How to use Eq. 14 to obtain the concurrence as function of temperature and magnetic field? More details are needed.

5. In line 127, the change of concurrence at low temperatures is presented. What about the high temperatures?

6. In line 244, it has been evidenced that these QCPs play a fundamental role in the work and efficiency of the proposed cycle. Related references?

Comments on the Quality of English Language

No

Author Response

Dear Referee, we appreciate your criticism, which has allowed us to improve our work.

Please see the attached document,

Best regards,

Francisco J. Peña.

Reviewer 2 Report

Comments and Suggestions for Authors

The authors investigate the efficiency of a quantum engine comprised of two qubits. They explore this efficiency as a function of field anisotropy. They show that adjusting the anisotropy can extend the range of ideal efficiency in the engine. They recover the efficiency of previous work in the appropriate limit.

This is a nice paper, and should be published. However, I have some minor suggestions that the authors should consider.

I am surprised that the authors do not reference the work of Sebastian Deffner, e.g. Symmetry 13 (6), 978 (2021).

I find it difficult to read the axes labels in Fig. 1. Figures 2 and 3 are better, but still not ideal. I recommend making the axes-label fonts larger and darker.

Equation (8) gives “specific heat” as C=dU/dT. Assuming the internal energy is an extensive quantity, this would be the heat capacity, not specific heat. Please check.

After the authors have considered these suggestions, I recommend publication in Technologies.

Comments on the Quality of English Language

The written English is fine. I have no specific suggestions.

Author Response

(The authors gave the same response as above.)
